# Simulating the structural phase transitions of metal-organic frameworks with control over the volume of nanocrystallites

Larissa Schaper[1] & Rochus Schmid [1✉]

Flexible metal-organic frameworks (MOFs) can undergo structural transitions with significant pore volume changes upon guest adsorption or other external triggers while maintaining their porosity. In computational studies of this breathing behavior, molecular dynamics (MD) simulations within periodic boundary conditions (PBCs) are commonly performed. However, to account for the finite size and surface effects affecting the phase transition mechanism, the simulation of non-periodic nanocrystallite (NC) models without the constraint of PBCs is an important alternative. In this study, we present an approach allowing the analysis and control of the volume of finite-size structures during MD simulations by a tetrahedral tessellation of the (deformed) NC's volume. The method allows for defining the current NC's volume during the simulation and manipulating it regarding a particular reference volume to compute free energies for the phase transformation via umbrella sampling. The application on differently sized DMOF-1 and DUT-128 NCs reveals flexible pore closing mechanisms without significant biasing of the transition pathway. The concept provides the theoretical foundation for further research on flexible materials regarding targeted initialization of the structural phase behavior to elucidate the underlying mechanism, which can be used to improve the applications of flexible materials by targeted controlling of the phase transition.

[1] Ruhr-Universität Bochum, Faculty of Chemistry and Biochemistry, Computational Materials Chemistry Group, Universitätsstr. 150, 44801 Bochum, Germany.
✉email: rochus.schmid@rub.de

One peculiar feature of metal-organic frameworks (MOFs), which differentiates them from other porous systems like zeolites, is the ability of specific systems to undergo significant volume changes upon guest molecule adsorption or other external triggers like hydrostatic pressure, temperature, or magnetic fields[1,2]. This "breathing effect" has attracted great interest because of its potential use in various areas, like improving the working capacity in gas separation[3].

In this context, the so-called "negative gas adsorption" was recently reported for DUT-49 and related MOFs, where the crystal cell volume is reduced at a limiting pressure, reducing the gas loading at higher pressure[4,5]. All these breathing and gate-opening effects are accompanied by a structural transition and deformation of the MOF, which is possible due to the flexibility of the organic linkers or their connection to the inorganic secondary building blocks (SBUs). These 3rd generation MOFs are called soft porous materials[6]. Although these breathing phase transitions are intensively investigated both experimentally and by theoretical methods, the phenomenon still needs to be fully understood, and a prediction of, e.g., limiting gate pressures is still difficult[7]. Recently, experimental evidence for a dependence of the breathing effect on the crystallite size has been found, indicating an influence of the system-to-volume ratio[8]. Also, the influence of defects and disorder on the flexibility of MOFs is still being determined[9,10]. From a theoretical point of view, most studies employ artificial periodic boundary conditions (PBCs) to simulate crystalline materials with relatively small unit cells and ideal, non-defective MOF systems leading to a well-defined cell volume, which allows pressure control by barostats[11]. It has been recently shown that large unit cells and a numerically effective GPU-based simulation approach allow studying the first-order phase transitions with the presence of both phases within PBCs[12,13]. These meso-sized systems have a crystal size of ~100 nm side length.

In contrast to such PBC simulations, the structural phase transition of DMOF-1 and derivatives was investigated as non-periodic nanocrystallites (NCs) with an approximated surface termination model[14,15]. The simulations confirm a first-order phase transition behavior with an interface between the open and the closed pore phase moving through the system. Already for the small NCs of ~1 nm side length, size effects on the free energy barrier could be observed, indicating a surface hindrance on the phase transition. This would explain that smaller crystallites typically are less flexible or do not switch in experimental studies. The current results corroborate that the simulation of NCs beyond PBCs is a necessary route to fully reveal the atomistic details of the structural phase transition of MOFs. However, the

approach is hampered by two difficulties: (i) An appropriate model for the surface termination of the NC is needed, and (ii) the application of pressure and the control of the system's volume of an arbitrarily deformed NC is complex.

The surface termination of a MOF, which also represents a defect concerning the bulk system, needs to be better investigated in experimental and theoretical studies. However, assuming a saturation of dangling bonds by modulators, often used in synthesis, allows staging an approximate and idealized surface termination[16,17]. More problematic is the second point, namely that a clear definition of the NC's volume needs to be included in contrast to PBC simulations. In this context, it has to be considered how hydrostatic pressure can be typically simulated. In experiments, hydrostatic pressure acts as a force on the surface of the crystal exerted by a medium, i.e., the surrounding gas, which is adsorbed simultaneously. Therefore, pressure is connected to adsorption. In a theoretical simulation, disentangling this can be advantageous since pressure can be simulated in the absence of an adsorbing gas as a force acting on the cell parameters. As a result, the system's volume is always well-defined by the unit cell. In a PBC simulation, there is no surface on which a medium could act. In contrast, there is no such well-defined volume for a deformed NC during a phase transition.

Figure 1 gives an overview of our various concepts to handle finite-size systems during simulation. In the NC simulations of MOF NCs, a distance restraint between SBUs on the edges of the system was used to exert a mechanical force on the system, mimicking the effect of pressure and inducing the structural phase transition[14]. This mechanical force has, of course, nothing in common with hydrostatic pressure and is derived from the expected reaction coordinate, at the same time enforcing a particular mechanism of pore closing.

To alleviate this bias, we recently considered the pressure bath method, where a fictitious medium is added, actually exerting pressure on the surface of the NC[18]. This method was inspired by the experimental mercury porosimetry, where a non-wetting fluid like mercury is used, which cannot enter the MOF nanopores. Therefore, in the absence of adsorption, the compression of a MOF crystallite could be experimentally triggered by pressure. In the simulation, spherically large particles that strongly attract themself but interact only repulsive with the MOF are used, leading to considerable surface tension. Using a PBC simulation cell with the NC embedded in the pressure bath medium, the structural phase transition could be investigated in a constant pressure (NPT) ensemble. In contrast

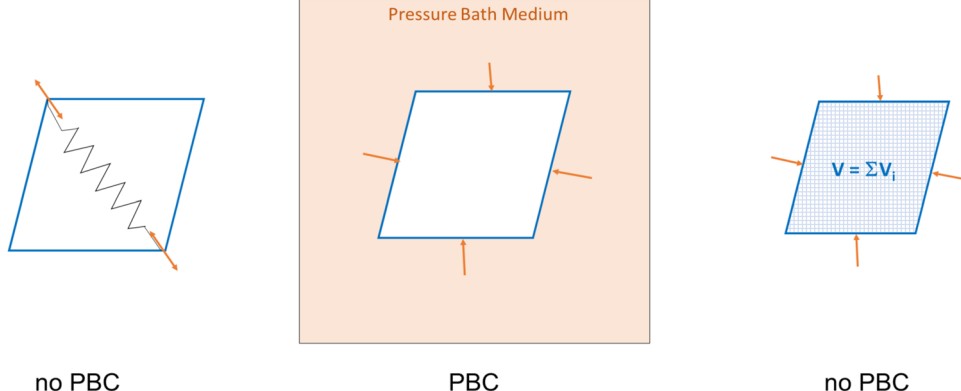

**Fig. 1 Handling of finite size systems in MD simulation.** The first approach introduces a mechanical force to initialize the pore closing mechanism by lowering the distance between opposite-corner SBUs. The pressure bath concept returns to PBCs: A finite-sized system is set up in a fluid. This system as a whole is simulated using PBCs. The third approach, introduced in this study, focuses on the volume of the finite size system, i.e., to get access to the pore volume and influence it.

to the previous simulations with a fictitious mechanical force, various mechanisms could be observed, indicating the absence of a bias on the pore closing path.

Despite the intriguing similarity of the pressure bath method to the actual experimental conditions, the method has several drawbacks. First, the free volume of the sufficiently large simulation cell must be densely packed with the pressure bath medium, adding a substantial numerical overhead to the simulation since the overall number of particles is substantially increased. Furthermore, simulations of negative pressure for reopening a system are impossible due to the purely repulsive potential between the pressure bath and MOF. Because of the high surface tension, first pore closing events are observed at the corners and edges of the NC, where the curvature of the NC surface is highest. This is a realistic effect, but it is possibly artificially exaggerated for the case of very small NCs, as in the simulations. Most importantly, however, due to significant pressure fluctuations in the pressure bath itself, the phase transition of the NC is not triggered at a fixed global pressure, and integration of the free energy is not possible.

Due to these drawbacks, a general approach is ultimately needed in order to access the volume of MOF NCs and to be able to manipulate it during the simulation while keeping the flexibility of the pore closing pathway, as observed in the pressure bath studies. A sufficiently accurate approximation of the deformed unit-cell volume is necessary to do this for an arbitrarily shaped NC. We propose to use a tetrahedron 3D tessellation of the individual pore and the summation of the individual pore volumes to achieve this. The following section discusses the general approach of the volume control method, including the necessary forces resulting from a harmonic restraining potential term. The approach is demonstrated for two different MOF NCs with the underlying topology **pcu**, namely the DMOF-1 (Zn$_2$(bdc)$_2$(dabco))[19] and DUT-128 (Zn) (Zn$_2$(4,4'-bpdc)$_2$(dabco); 4,4'-bpdc: 4,4'-biphenyldicarboxylate)[20] investigated already in previous studies as finite size systems.

## Methodology

**A collective variable for the NC volume**. In order to overcome the problem that the NC's volume is not well-defined by the simulation cell as in PBC, the individual pores of finite-size MOF NCs are described by tetrahedra to calculate the volume and its derivatives. To clarify this strategy, a MOF pore is decomposed to its geometrical basics.

Figure 2 shows the DMOF-1 unit-cell corresponding to only one pore of the NC. The pore is defined by the positions of eight paddle-wheel units at the corners and can be treated as a polyhedron. The corners (black dots in Fig. 2) represent the vertex positions of the paddle-wheels connected by the linkers (black lines). However, if deformed pores are considered, the geometrical description of an orthorhombus is not sufficient, which is why the volume has to be determined by a more suitable geometric object. In order to develop a general approach, which can be used for an arbitrary pore shape and, therefore, for different MOFs, the unit-cell is divided into five tetrahedrons as shown in Fig. 2. There is one tetrahedron inside the pore (colored in blue and orange, respectively) and four tetrahedrons built by the sides of the pore and one of the faces of the centered tetrahedron. Four paddle-wheel vertices define each tetrahedron. There are two possibilities to insert these five tetrahedrons. For an orthorhombic pore, both possibilities are equivalent, but for a deformed system, the resulting volume depends on how the tetrahedrons are placed. To avoid this ambiguity, both possibilities are considered and the resulting total volumes are averaged.

The volume of one tetrahedron with the vertices O, A, B, and C can be calculated by

$$V_{tet} = \frac{1}{6} \cdot \left| \vec{a} \cdot \left( \vec{b} \times \vec{c} \right) \right|$$
$$= \frac{1}{6} \cdot \left| a_x \cdot (b_y \cdot c_z - b_z \cdot c_y) + a_y \cdot (b_z \cdot c_x - b_x \cdot c_z) + a_z \cdot (b_x \cdot c_y - b_y \cdot c_x) \right|,$$
(1)

with the vectors $\vec{a}$, $\vec{b}$, and $\vec{c}$, pointing from O to A, B, and C, respectively. Note that labeling the vertices is arbitrary and must be done once in the beginning for each of the five tetrahedrons, forming one of the two representations.

The derivative of the volume with respect to the Cartesian coordinates of the vertices can be determined via the derivatives with respect to the components of the vectors $\vec{a}$, $\vec{b}$, and $\vec{c}$.

$$\frac{\partial V}{\partial a_x} = \frac{1}{6} \cdot \left( b_y \cdot c_z - b_z \cdot c_y \right)$$
$$\frac{\partial V}{\partial a_y} = \frac{1}{6} \cdot \left( b_z \cdot c_x - b_x \cdot c_z \right)$$
$$\frac{\partial V}{\partial a_z} = \frac{1}{6} \cdot \left( b_x \cdot c_y - b_y \cdot c_x \right)$$
$$\frac{\partial V}{\partial b_x} = -\frac{1}{6} \cdot \left( a_y \cdot c_z - a_z \cdot c_y \right)$$
$$\frac{\partial V}{\partial b_y} = -\frac{1}{6} \cdot \left( a_z \cdot c_x - a_x \cdot c_z \right)$$
$$\frac{\partial V}{\partial b_z} = -\frac{1}{6} \cdot \left( a_x \cdot c_y - a_y \cdot c_x \right)$$
$$\frac{\partial V}{\partial c_x} = -\frac{1}{6} \cdot \left( b_y \cdot a_z - b_z \cdot a_y \right)$$
$$\frac{\partial V}{\partial c_y} = -\frac{1}{6} \cdot \left( b_z \cdot a_x - b_x \cdot a_z \right)$$
$$\frac{\partial V}{\partial c_z} = -\frac{1}{6} \cdot \left( b_x \cdot a_y - b_y \cdot a_x \right),$$
(2)

These derivatives are in turn used to compute the forces acting on the actual atoms of the paddle-wheels by the chain rule, since the vertices are defined as their COMs. The COM of a paddle-wheel is calculated by taking into account both metal atoms and the four carboxylates. For all vertices in the interior of the NC, the volume derivatives of the eight neighboring unit-cells will always exactly compensate. Thus, for the total volume derivative of the entire NC, it is sufficient to compute derivatives only for the vertices (paddle-wheel units) at the surface of the NC. For analysis, the volume is computed on a per unit-cell basis.

In order to control the individual pore volume, a volume restraining potential is applied to the MOF NCs, which modifies the acting forces by a volume-dependent harmonic part. The external potential has the form

$$U(\mathbf{r})_{ext} = \frac{1}{2} \cdot k \cdot \left( V(\mathbf{r}) - V_{ref} \right)^2,$$
(3)

where $k$ is the force constant, $V_{ref}$ is the reference volume, $V$ is the total volume of the NC, and $\mathbf{r}$ represents the position of the atoms in space. The quadratic term adds a penalty proportional to the deviation from the reference volume. Note, that only the atoms of the paddle-wheel unit (metal and carboxylate atoms) defining the tetrahedron vertices via their COM are contributing and are affected by forces from the restraint. The introduction of this external potential allows for defining a collective variable (CV) and performing umbrella sampling (US)[21,22] to get a free energy profile.

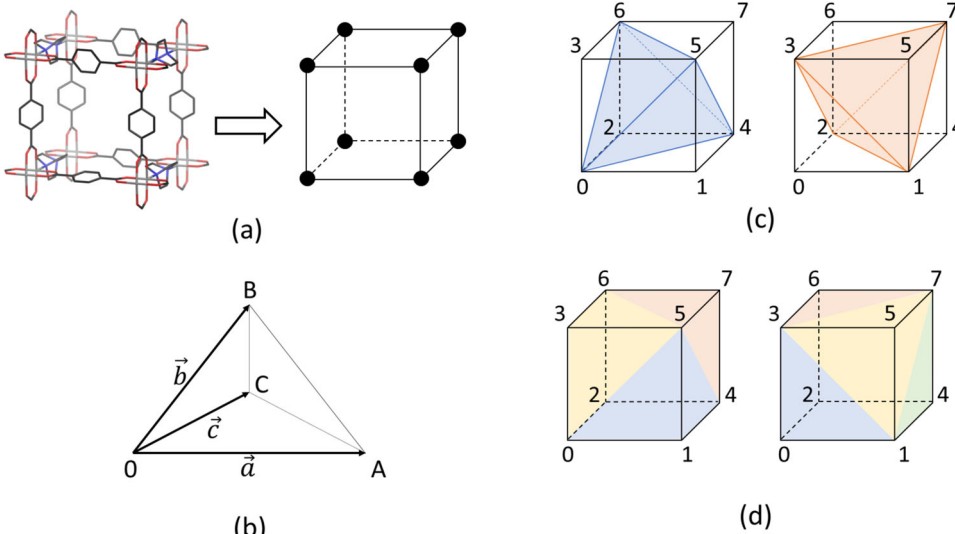

**Fig. 2 Geometrical consideration of porous materials. a** The smallest possible DMOF-1 NC with only one pore, shown without stubs and H atoms, can be treated as a cuboid, which is defined by the eight paddle-wheel vertices. **b** The cuboidal volume can be tessellated by tetrahedrons spanned by the vectors $\vec{a}$, $\vec{b}$, and $\vec{c}$. In order to describe an arbitrary pore shape, the pore volume can be obtained by the sum of the volume of five tetrahedrons: One inner tetrahedron (**c**) and four tetrahedrons built by the sides of the pore and one face of the inner tetrahedron (**d**). The two possibilities to tessellate the pore illustrated on left and right in (**c**) and (**d**).

**Computational details**. All force field calculations were performed with the LAMMPS molecular mechanics program package[23,24]. The volume constraint has been implemented in Python using the LAMMPS fix python/invoke in order to perform calculations of the NC's volume and the forces resulting from the external potential $U_{ext}$ (Eq. (3)) in each timestep. For numerical efficiency, the computation is done in parallel, distributing over the pores, and the core routines are accelerated by a just-in-time compiler Numba[25]. The implementation was tested to give energy conserving dynamics in a microcanonical simulation (see further details in Supplementary Note 1.2 and Supplementary Fig. 1).

The open pore form of the differently sized DMOF-1 and DUT-128 NCs was constructed by the reverse topological approach (RTA)[26–28]. The procedure of generating the cubic NCs is described in detail in reference[14]: First, the blueprints for the differently sized nanocrystallites were prepared by slicing the respective supercells of the **pcu** topology and extending them by vertices for the stubs. Then, the BBs were assigned to their respective vertices. The NCs were named after the number of paddle-wheel units in the x-, y-, and z-direction, e.g., the $5 \times 5 \times 5$ NC has five paddle-wheel units and four pores in each spatial direction. In this study, the $3 \times 3 \times 3$ up to the $6 \times 6 \times 6$ DMOF-1 NCs and the $3 \times 3 \times 3$ up to the $9 \times 9 \times 9$ DUT-128 NCs were investigated. The interactions of the DMOF-1 NCs were described by the first principles parameterized MOF-FF force field[29,30].

The structures were structurally optimized using the conjugate gradient minimization method and, in addition, the steepest descent algorithm to relax the cell. First, the atomic positions were optimized with an energy convergence criterion of $0.15\,\mathrm{kcal} \cdot \mathrm{mol}^{-1} \cdot \mathring{A}^{-1}$. Then, the cell and atomic positions were changed by the steepest descent algorithm until the forces acting are lower than $0.1\,\mathrm{kcal} \cdot \mathrm{mol}^{-1} \cdot \mathring{A}^{-1}$. Finally, the atomic positions were optimized once more with an energy convergence criterium of $0.05\,\mathrm{kcal} \cdot \mathrm{mol}^{-1} \cdot \mathring{A}^{-1}$.

The optimized structures were heated up from $10\,\mathrm{K}$ to $300\,\mathrm{K}$ within $50\,\mathrm{ps}$ in the NVT ensemble with a time step of 1 *fs*. The initial velocities were generated by a Maxwell-Boltzmann distribution. The system was coupled to the Nosé-Hoover chain thermostat[31] with a relaxation time of 0.05 ps. All subsequent simulations were carried out with a relaxation time of 0.1 ps and a temperature of 300 K, whereby the temperature is only a reference for the kinetic energy in the system.

To apply the volume restraining external potential, a proper choice of the force constant $k$ is needed since it controls the strength of the potential to bias the volume. For this purpose, a screening experiment was performed using DMOF-1 NCs in the op form as a test system in order to determine its magnitude. Further details are given in the Supplementary Note 1.1. Supplementary Table 1 lists the tested force constants and the difference between the mean volume of the respective NC during the simulation and the reference volume for the respective system. As a result, a force constant $k = 0.002\,\mathrm{kcal} \cdot \mathrm{mol}^{-1} \cdot \mathring{A}^{-6}$ was used in all further SMD calculations to maintain a difference between the mean volume of the respective NC during the simulation and the reference volume of approximately $\Delta V = 10\,\mathring{A}^3$ up to $\Delta V = 80\,\mathring{A}^3$ for all investigated sizes, while the fluctuation of the volume is $\Delta\Delta V \geq \pm 50\,\mathring{A}^3$.

The heated systems were first equilibrated for $100\,ps$ without any restraints. Then, the external potential for volume control was switched on. First, the reference volume was kept constant to the current volume for 10 ps. Then, the reference volume was linearly decreased from 1170 $\mathring{A}^3 \cdot n_{pores}$ to 610 $\mathring{A}^3 \cdot n_{pores}$ for DMOF-1 and from 2200 $\mathring{A}^3 \cdot n_{pores}$ to 8200 $\mathring{A}^3 \cdot n_{pores}$ for DUT-128, where $n$ refers to the number of pores in the respective NC, with a speed of 0.1 $\mathring{A}^3$ per time step. For the $5 \times 5 \times 5$ DMOF-1 NC, reversed calculations were performed, in which the reference volume was linearly increased from 610 $\mathring{A}^3 \cdot n_{pores}$ to 1170 $\mathring{A}^3 \cdot n_{pores}$ to simulate the reopening from the closed pore form to the open pore form.

The free energy needed to transform the differently sized NCs from the open to the closed pore form was determined by US simulations[32] using the volume as CV. For this purpose, snapshots were taken from steered Molecular Dynamics (SMD) simulation every $\Delta V_0 = 100\,\mathring{A}^3$ for the $3 \times 3 \times 3$ and the $4 \times 4 \times 4$ DMOF-1 NC, and every $\Delta V_0 = 200\,\mathring{A}^3$ for the $5 \times 5 \times 5$ and the $6 \times 6 \times 6$ DMOF-1 NC as well as for all DUT-128 NCs. For each window, a 100 ps simulation was carried out with a fixed reference volume

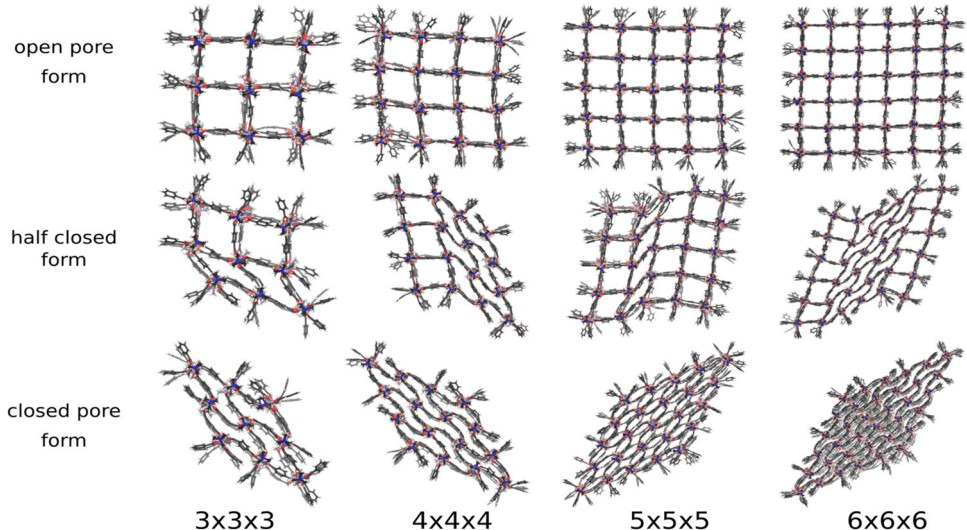

**Fig. 3 Structural transition of differently sized DMOF-1 NCs.** The open, partially closed, and closed pore forms are obtained by applying the extended volume restraining potential during simulation. All system sizes end up in a diamond-shaped closed pore form.

and a force constant of $0.001 \ \mathrm{kcal \cdot mol^{-1} \cdot Å^{-6}}$ for the $3 \times 3 \times 3$ and the $4 \times 4 \times 4$ DMOF-1 NC, and a force constant of $0.0001 \ \mathrm{kcal \cdot mol^{-1} \cdot Å^{-6}}$ for the $5 \times 5 \times 5$ and the $6 \times 6 \times 6$ DMOF-1 NC as well as for all DUT-128 NCs. The CV was recorded every 100 steps for DMOF-1 NCs and every 500 steps for DUT-128 NCs. The overlap of the respective windows was checked by visualization of the distributions of the CV (see Supplementary Note 1.3 and Supplementary Fig. 2 for DMOF-1 as well as Supplementary Note 2.1 and Supplementary Fig. 4 for DUT-128).

The free energy profile was determined by the Weighted Histogram Analysis Method (WHAM) using the wham code[33] and a convergence threshold of $0.0001 \ \mathrm{kcal \cdot mol^{-1}}$. For each system size, three independent runs of SMD and consecutive US calculations with different initial conditions were carried out and the resulting free energies were averaged.

## Results and discussion

**Phase transformation of the model system DMOF-1.** The volume restraining potential is applied to DMOF-1 NCs to initiate the pore closing mechanism from an open to a closed form via SMD. As previously observed for the pillared-layer MOF NCs in the pressure bath medium, the pore closing propagates through the x-y-plane for all z-layers of the NC. Hence, the process is again analyzed within this plane. In the following, the phase transition pathways of DMOF-1 NCs with the sizes $3 \times 3 \times 3$ up to $6 \times 6 \times 6$ are discussed concerning their flexibility to end up in the closed pore form.

Figure 3 shows the open and closed pore, as well as selected intermediate structures. All investigated systems end up in the diamond-shaped closed pore form. Transition structures with one phase boundary are the most probable intermediate structures, but also, two phase boundaries are observed during the phase transformation. To discuss the transition path in detail, the individual pore sizes are averaged along the z-axis to show the view on the xy-plane (Fig. 4). The center of the pores are colored according to the respective volume. Blue-colored dots represent closed pores with a volume of ~610 Å³, whereas a red-colored symbol indicates open pores with a volume of ~1170 Å³.

The closing mechanism of the $3 \times 3 \times 3$ NC and the $4 \times 4 \times 4$ NC is the archetypical mechanism for the phase transformation: The process starts at one corner and propagates asymmetrical through the NC, i.e., the pore closing occurs in one direction first, which induces the closing of the neighbored layer, and ends up in

a diamond-shaped structure of the entire NC. The $5 \times 5 \times 5$ and $6 \times 6 \times 6$ NC show a slightly different transition mechanism: The (centered) inner layer closes first, whereas no outer layers or corners are affected initially. For the $5 \times 5 \times 5$ NC, a reopening of closed pores is observed after one of the outer layers is entirely closed. Then, the known layer-by-layer pore closing proceeds, ending in the diamond-shaped closed pore form. In the case of the $6 \times 6 \times 6$ NC, the central and a neighboring layer close first, resulting in two phase boundaries. The fact that the outer layers are still open, whereas the mid-layers are entirely closed, causes structural tension in the NC and explains the reopening of one of the middle layers and the spontaneous pore closing of the outer layer at $t = 480$ ps. In the end, the diamond-shaped structure is formed again by a layer-by-layer closing.

These pore closing mechanisms are selected results from several SMDs of each investigated system. They demonstrate trends and allow for testing of the volume control approach. However, the SMD settings, like the speed of change of the reference volume or the chosen force constant, influence the individual processes. In addition, they depend on the initial conditions, and in order to elucidate the pore closing mechanism as statistically meaningful, more than three performed SMD simulations for each system size would be needed.

Nevertheless, the results demonstrate that the method maintains the flexibility to correct a (partially-)closed pore form to a more favored one. This allows two conclusions: First, the chosen force constant seems suitable to fix the system in a given conformation on the potential energy surface, but it allows enough fluctuation in the volume of the NCs to reopen closed pores. Second, it is validated that only one phase boundary in the transition structure and the diamond-shaped closed pore form are the favored formations of the NCs.

*Umbrella sampling calculations of the model system DMOF-1.* In the next step, the approximate total volume of the NCs, determined by the tetrahedron tessellation, was chosen as a CV to compute the Helmholtz free energy profile of the phase transformation by US. The initial structures for the US windows were obtained from snapshots of the SMD discussed in the previous section. Hence, e.g., for the $5 \times 5 \times 5$ NC and the $6 \times 6 \times 6$ NC, some windows start with the transition structures containing two phase boundaries. Interestingly, it is found that these structures reopen during the simulation with a constant reference volume

**Fig. 4 Pore size analysis of differently sized DMOF-1 NCs.** The pores are colored according to their size. The pore closing is induced by the harmonic restraining potential. For the $3 \times 3 \times 3$ and the $4 \times 4 \times 4$ NC, the common layer-by-layer pore-closing mechanism is observed. In contrast, the pore-closing mechanism of the $5 \times 5 \times 5$ and the $6 \times 6 \times 6$ NC starts at the centered layers and propagates to the edges.

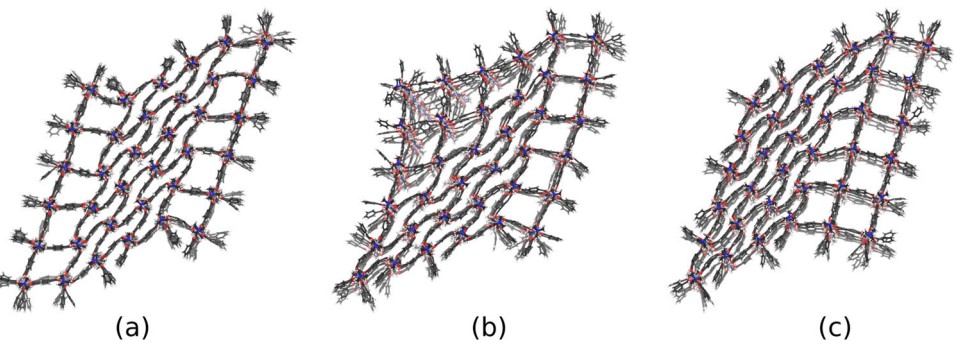

**Fig. 5 Chronological snapshots from the trajectory of the $6 \times 6 \times 6$ NC with the reference volume of $V_0 = 1 \times 10^5$ Å$^3$.** **a** $t = 0$ ns, **b** $t = 0.03$ ns, and **c** $t = 0.1$ ns. While the reference volume is kept constant, the system closes the outer layer pores (from (**a**) to (**b**)) and reopens its inner pores (from (**b**) to (**c**)), avoiding two phase boundaries present in (**a**).

and transform into a configuration with only one phase boundary, which is expected to reduce the strain energy resulting from such a phase boundary.

Figure 5 shows snapshots from the trajectory of the $6 \times 6 \times 6$ NC, biased by a reference volume of $V_0 = 1 \times 10^5$ Å$^3$, as an example. In contrast, a structure with only one phase boundary does not reopen during the simulation, corroborating that structures with one phase boundary are energetically preferred.

Figure 6 displays the averaged free energy profiles for the phase transition of all four investigated system sizes determined from the US calculation concerning the volume restraint.

In Fig. 6a, the total free energies are plotted against the RC. All curves have roughly the same shape. In all cases, the energetically lowest structure is the open pore form, which has been taken as zero. When the total volume of the NC is decreased, the energy rapidly increases until it reaches a local maximum. For small volumes, the energy curve has a local minimum representing the closed pore form of the NC of the respective size. The free energy profiles allow two essential conclusions: First, for all sizes, the phase transformation implies overcoming a free energy barrier between the open and the completely closed pore form, which is substantially smaller than for the PBC case[14,34] after normalizing per pore (see

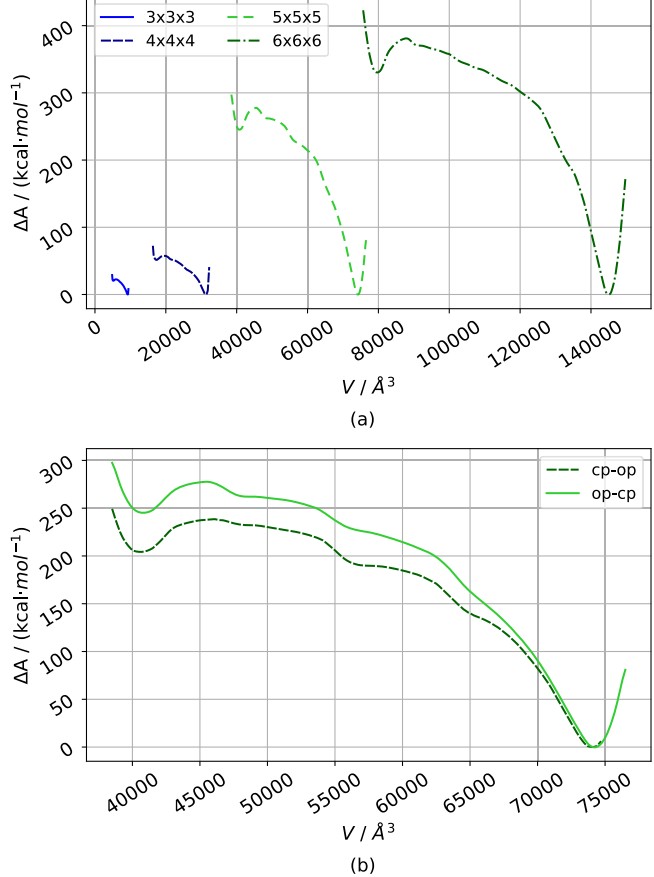

**Fig. 6 Free energy curves for DMOF-1. a** The total Free energy $\Delta A$ curves of the structural phase transition of the differently sized DMOF-1 NCs reveal that the open pore form is lower in energy. **b** The free energy $\Delta A$ profile of the $5 \times 5 \times 5$ NC based on US simulations using initial structures obtained by an SMD from the open to the closed pore form (dark green) and from the closed to the open pore form (light green) has a similar shape.

Supplementary Note 1.4 and Supplementary Fig. 3). Second, the closed pore form is an energetic minimum structure of the DMOF-1 NCs. Thus, the calculations based on MOF-FF predict a stable structure for the closed pore form. However, it is substantially higher in energy than the open pore form, meaning that the open pore form is the more favored structure. Experimentally, DMOF-1 is a non-flexible MOF, which tends to amorphize under hydrostatic pressure[34]. However, bond breaking can not be simulated for a non-reactive model like MOF-FF, and a fictitious stable but high-energy closed pore form is observed in the simulations.

To validate the RC chosen for the US approach and to check convergence, the SMD to obtain the initial structures was repeated for the $5 \times 5 \times 5$ NC. In contrast to the initial run, ramping the reference volume up enforced the reversed transition from the closed to the open pore form. In Fig. 6b, both free energy curves obtained via SMD in opposite directions are given, showing a slight deviation. This indicates the remaining statistical error due to insufficient convergence.

However, the shape of the two curves is similar with an energetic minimum at approximately $4 \times 10^4$ Å$^3$ for the closed pore form and at approximately $7.3 \times 10^4$ Å$^3$ for the open pore form. Thus, the method gives qualitatively correct results, but an even longer simulation time would be needed to improve the quantitative results.

When comparing energy profiles of different system size, it is evident that both the energy barrier and the difference between

open and closed form becomes higher when the system size is increased. A similar trend was observed for the unnormalized free energy curves in reference[14], where a simple mechanical force initiate the phase transition. This was explained by the increasing total barrier due to the larger number of pores, which have to be closed simultaneously, which also appears valid for the here presented results. For even larger NCs, the barrier is expected to converge to a particular value since only a limited fraction of the pores will have to undergo a structural transition for the pore closing. This will allow us to extrapolate the limiting barrier for arbitrarily large systems.

This proof-of-concept study demonstrates that US using the NC's total volume as a CV results in a similar transition pathway as the US using a restrained distance between SBUs on opposite edges of the NC[14]. Since the new volume-based CV allows for more flexibility concerning the pore closing and reopening, it can be concluded that this represents a realistic scenario for the mechanism of the phase transformation. In addition, the proposed method solves the problems of the pressure bath method, e.g., the reversed structural transition from closed to open pore form can be mechanistically investigated.

**Impact of an Extended Linker: DUT-128 NCs.** In order to generalize the volume control method, it is also applied to DUT-128, which has an experimentally proven stable closed pore form. The DUT-128 NCs are related to DMOF-1, i.e., they have the same underlying topology but the linker in x- and y-direction is extended, leading to a larger dispersive interaction in the closed form. In the first step, the DUT-128 NCs with the sizes $3 \times 3 \times 3$ up to $9 \times 9 \times 9$ were investigated. Similar to DMOF-1 NCs, the pore closing is triggered via SMD and observed along the x-y-plane for all z-layers of the NCs.

Figure 7 shows the open and closed pore form, as well as selected transition structures. Most of the investigated systems end up in the diamond-shaped closed pore form. The transition structures for the $3 \times 3 \times 3$ and the $6 \times 6 \times 6$ up to the $9 \times 9 \times 9$ NC reveal that the phase transition occurs subsequently via propagation of a one phase boundary through the system from one edge to the opposite one. This is the dominant closing mechanism for all investigated systems and sizes.

In Fig. 8a, several pore closing mechanisms of the $6 \times 6 \times 6$ NC are displayed from individual SMD simulations. The most significant differences are visible at the beginning of the process. The formation of one phase boundary, propagating through the system, can be initialized from different sides and edges of the NC. Despite the different orientations of the phase boundary and the consecutive direction of the propagation, they end up all in the same diamond-shaped structure.

An exception is observed in the trajectories for the $4 \times 4 \times 4$ NC: Figure 8b reveals that the partially closed pore form differs in one case. Here, no distinct phase boundary can be identified, and the pore closing starts randomly in several pores, resulting in a non-diamond-shaped form. This demonstrates that the volume control method does not enforce a specific mechanism and allows to explore the potential energy landscape of the structural change mechanism in an unbiased way.

*Umbrella sampling calculations of DUT-128 NCs.* Due to the enlarged linkers, the number of atoms in the systems increases compared to DMOF-1 NCs with an equal number of pores. This results in a larger numerical effort especially for the US simulations, and thus we focused on the NCs up to a size of $5 \times 5 \times 5$.

Figure 9 shows the total free energy curves of the pore closing mechanism of the $3 \times 3 \times 3$ up to the $5 \times 5 \times 5$ DUT-128 NCs. As expected and before observed for DMOF-1 NCs, they reveal a

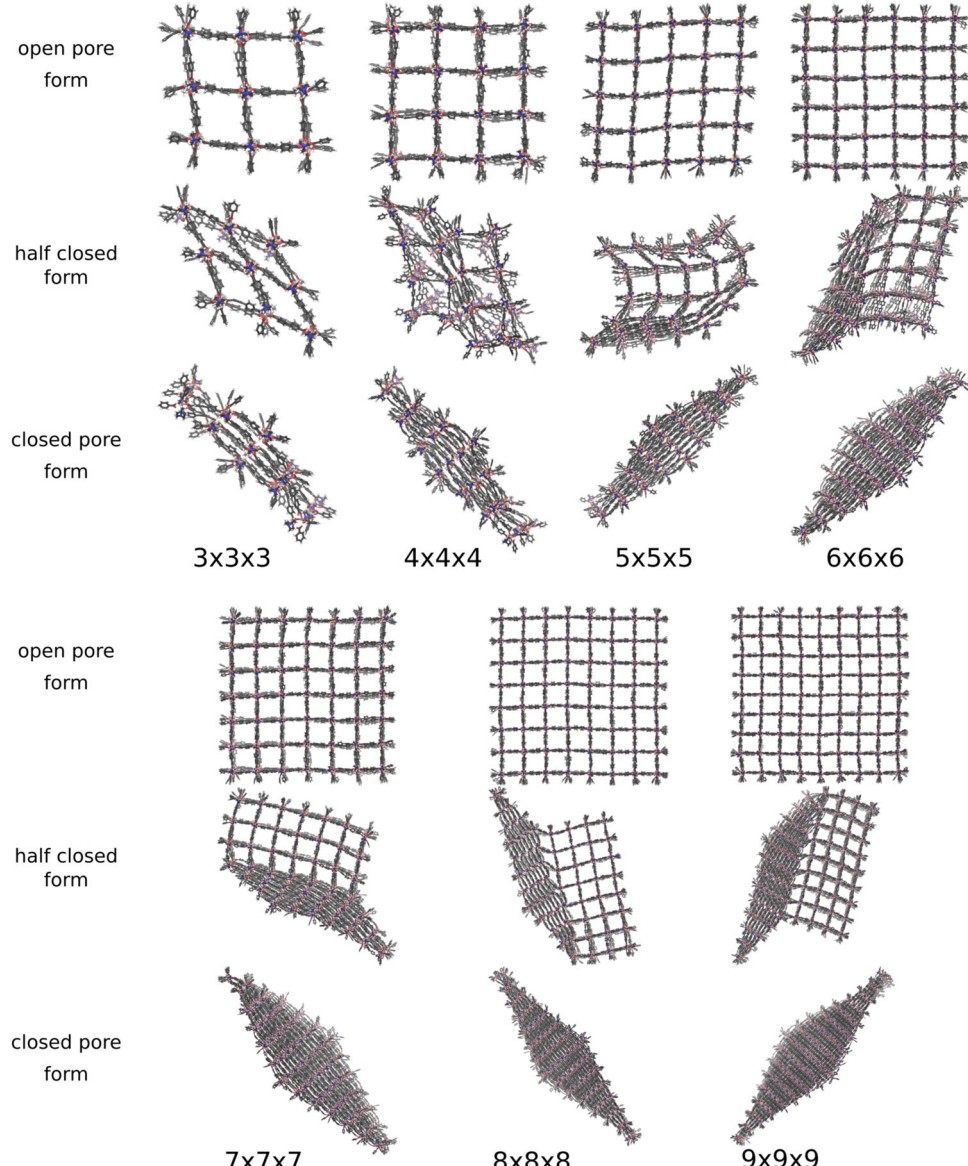

**Fig. 7 Structural transition of differently sized DUT-128 NCs.** The open, partially closed, and closed pore forms are obtained by applying the extended volume restraining potential during simulation. Although the transition structures are different, all systems end up in a diamond-shaped closed pore form.

substantial size effect, i.e., the energy difference between closed and open pore form, as well as the free energy barrier between the states, is increasing for larger system sizes. A plot of the free energy normalized per pore can be found in Supplementary Fig. 3 (Supplementary Note 1.4).

However, in contrast to DMOF-1, for DUT-128, the closed pore form is now lower in energy than the open pore form due to the stabilizing dispersive interactions for the biphenyl bridge in the linker. This also confirms the experimentally observed stable closed pore form of DUT-128. It is instructive to compare the energetics of phase transformation for the $5 \times 5 \times 5$ NCs of DMOF-1 with DUT-128: For the smaller DMOF-1, the closed pore minimum is about 250 kcal · mol$^{-1}$ above to open form, but with a small barrier of only about 25 kcal · mol$^{-1}$, whereas the larger DUT-128 the closed form is more stable by about 150 kcal · mol$^{-1}$ and with a substantially larger barrier of about 75 kcal · mol$^{-1}$. Thus, also the activated open pore form of DUT-128 can be observed. Note that under experimental conditions, phase transformations are usually triggered by guest molecule adsorption, which will alter the free energy curves. In our

computational experiment, however, we are able to observe the process disentangled from adsorption and focus on the properties of the pure MOF NC.

A further feature of the free energy curves is that the local minima in the plateau region between the closed and open pore form increase in number and depth with growing system size. For DMOF-1, they are much less pronounced. Each minimum corresponds to an entirely closed row of pores. For the $5 \times 5 \times 5$ NC with four rows of pores, exactly three such minima are observed before the system is closed. Such minima for the intermediate system with phase coexistence have previously been observed for simulations in PBC when large supercells are used[12,13].

## Conclusion

The methods for computational investigation of structural phase transitions of MOFs beyond PBCs using NCs have been extended by developing and testing an approach to control the volume of irregularly shaped objects accurately. To determine the volume of an arbitrarily shaped pore, it is tessellated by tetrahedrons, whose

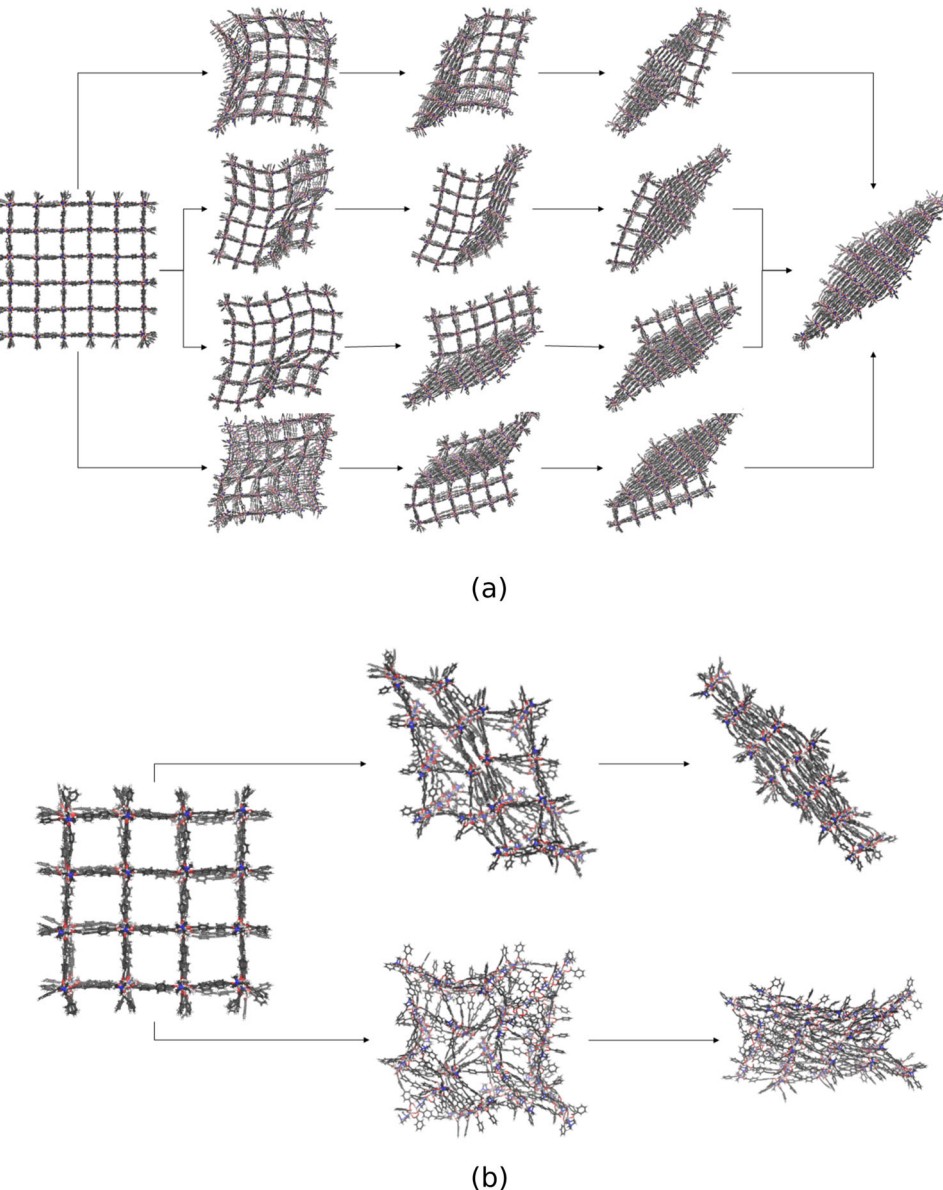

(a)

(b)

**Fig. 8 Structural phase transition pathway of DUT-128 NCs with different sizes. a** For the 6 × 6 × 6 NC, all phase transitions end up in the diamond-shaped closed pore form. Significant differences during the phase transition are observable at the beginning of the mechanism. For smaller reference volumes, a phase boundary between open and closed pore forms travels through the system, independent of how the pore closing starts. **b** For the 4 × 4 × 4 NC, a different closed pore form than the diamond-shaped one is observed.

volumes are calculated individually and summed up. The procedure is repeated for every pore of the NC. A volume restraining potential is employed to control the NC's total volume without enforcing an individual pore volume and to avoid any bias on the phase transformation mechanism.

The volume control method was applied to differently sized DMOF-1 and DUT-128 nanocrystallites to investigate their structural transformation and compare it to the formerly obtained pressure bath results[18]. For this purpose, MD simulations on the FF level of the structural transition from the open to the closed pore form were performed. The non-synchronous phase transformation results in differently shaped closed pore forms revealing various possible transition pathways, especially for larger-sized linkers. The results agree with those obtained previously by invoking hydrostatic pressure on the NC by the pressure bath method, demonstrating that direct volume control allows similar flexibility to explore the phase transition

mechanism without bias. In addition, the direct volume control is a suitable CV and can be used as a bias potential to perform US to calculate the free energy profiles of the structural phase transition. In contrast to DMOF-1, for the DUT-128 NCs with an extended linker, the closed pore form is computed to have lower free energy than the open pore form. However, the latter should also be a metastable phase because of a larger barrier.

The approach has three main advantages compared to the pressure bath method. First, it requires a substantially lower numerical effort because the pressure mediating particles are not needed. Second, it solves the main problems of the pressure bath method, namely that the pressure fluctuations in the pressure bath prevent the determination of pressure-volume curves and, thus, the computation of free energy curves. Moreover, third, the reversed process of a structural transition from the closed to the open pore form can also be investigated. Especially for the simulation of substantially larger NCs, e.g., with different aspect

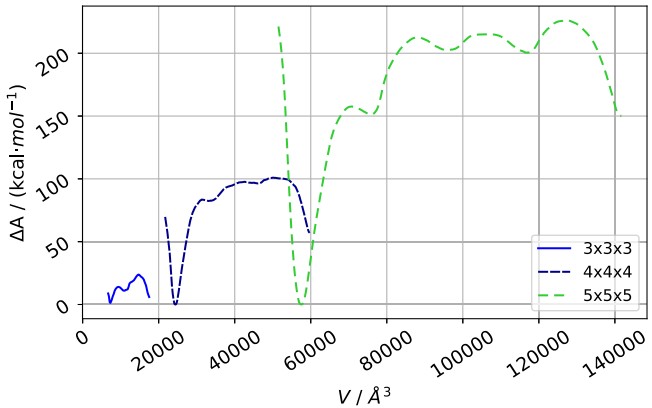

**Fig. 9 Free energy curves of DUT-128.** The total Free energy $\Delta A$ curves of the structural phase transition of the differently sized DUT-128 NCs reveal that the closed pore form is lower in energy.

ratios, the direct volume control method will make such investigations computationally more tractable since size effects are observed experimentally for synthesized MOF powders, whose sizes are still one or two orders of magnitude larger than the here studied systems. In addition, the method could be combined with the explicit simulation of guest molecules surrounding the NC, which allows to study transport and adsorption in a joint fashion.

The results illustrate that the phase transition of MOFs can follow various possible pathways, which can only be simulated properly in the absence of the constraint of periodic boundary conditions. Consequently, it requires appropriate methods to sample the phase space, especially in low probability regions. In this context, the direct volume control is the ideal collective variable to study such processes. Its application to MOF NCs will lead to a more detailed understanding of the underlying mechanism, i.e., to elucidate the surface and transport effects of guest molecules inducing structural transition processes by determining the limiting pressure of closing, which can be used to improve the applications of flexible materials by the targeted design of the phase transition behavior.

## Data availability

Additional simulation details are provided in the Supplementary Information. The simulation input files generated for the study can be found in the github repository: https://github.com/cmc-rub/supporting_data/tree/master/109-Schaper-commschem-2023.

## Code availability

The Python code for the volume restraint can be found in the github repository: https://github.com/cmc-rub/supporting_data/tree/master/109-Schaper-commschem-2023.

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

## Acknowledgements
This work has been funded by the Deutsche Forschungsgemeinschaft (DFG) by grants (SCHM1389/10-1 and SCHM1389/10-2) within research unit FOR2433.

## Author contributions
R.S. designed the model and carried out the implementation. L.S. performed the calculations and analyzed the data. Both authors wrote the manuscript.

## Funding

## Competing interests
The authors declare no competing interests.

## Additional information

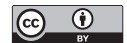

