## [Peer Review File · Communications Chemistry]

Reviewers' comments:

Reviewer #1 (Remarks to the Author):

In „Simulating the Breathing Phase Transitions of Metal-organic Frameworks : Volume Control of Nanocrystallites” Schaper and Schmid continue previous molecular dynamics (MD) investigations on framework flexibility on finite-sized crystals of metal-organic. The current work addresses the well-known DMOF-1 structure as a starting model and applies the methodology of direct volume control. The methodology is then applied to the recently reported framework material with extended ligand length named DUT-128. The application of direct volume control provides full access to the phase change and processes occurring upon framework deformation as well as the underlying free energy landscapes. This allows to sample nanocrystals of MOFs without periodic boundary conditions up to volumes of 1400000 Å³ (6x6x6 supercells). The manuscript is well written and illustrated. I particularly enjoyed the discussion on comparing the various computational approaches with respect to experimental procedures. The recent experimental observations of such finite crystal size phenomena can now finally be accessed computationally with even higher accuracy and lower costs which is a big step forward – Congratulations to the authors! As far as I can tell the manuscript and ESI contains all relevant information (github repository link will be provided when submitting a final manuscript for publication.) to reproduce the study and potentially extend it to other systems. As such I strongly recommend publication of this manuscript and in the following provide a few suggestions and questions that might improve the manuscript even further.

I suggest to add a figure to extend Fig 6 and 9 or a new panel with normalized (to a certain volume/energy) energy landscapes and directly compare as an overlay the shape of the free energy landscape upon increasing the size of the NCs.

Fig.8: a) and b) should be defined

Have the authors considered extracting information about the nature and impact of volume to surface ratios in this study? Is there a way to extract such information in particular on the formation of metastable states?

An open question in the community of flexible MOFs is the correlation of the activation energy for a transition determined with PBCs (usually per unit cell) to the activation barrier in a finite crystal. Do the authors see a way to extrapolate the current findings towards even larger super cells without actually having to perform the MD simulations?

Can this study be conducted with guest molecules included in the pores and would that be an accurate way of sampling guest-induced transitions similar to MD-GCMC methods?

Reviewer #2 (Remarks to the Author):

This manuscript reports on molecular dynamics simulations of structural transformations of MOF frameworks. The authors have been working on this topic using computer simulations, and this manuscript is one of them. The focus of this study is to improve a simulation technique and solve some problems confirmed in the pressure bath concept that the authors previously proposed. In this paper, the authors proposed the volume control method and successfully reproduced structural transformation processes from open pore form to closed pore form. However, the results obtained in this study basically followed those obtained from previous simulations using the pressure bath concept. In that sense, the novelty of this study is doubtful from a viewpoint of MOF chemistry and materials. Thus, this manuscript is not suitable for the publication in Communications Chemistry, and rather more suitable for a journal dealing with simulation techniques and theoretical approaches. Another comment is on the title of this manuscript. Considering the present results in which guest molecules are not treated in the simulations, "breathing phase transitions" should be replaced with "structural transitions" because breathing is accompanied by host framework transformation and guest molecule adsorption/desorption.

Reviewer #3 (Remarks to the Author):

The authors have produced an interesting piece of work on modelling the breathing behaviour of metal-organic frameworks via finite sized simulations of nano crystallite models. I believe this is an underdeveloped area in the field, firstly that the periodic models used to simulate these effects rarely agree with experimental observations, and finite sized models have only been recently introduced into the literature in the past 3-4 years. Therefore I welcome this new approach.

The authors begin with outlining how their new approach to modelling finite sized systems via tetrahedra and calculate the volume and derivative. Here I have a few questions relating to the terminology describing the shape of DMOF-1.

DMOF-1 is tetragonal in symmetry, the initial assumption in the text of Fig 2 it is stated " can be treated as a cube defined by the eight paddle-wheel vertices", would it not be treated as a cuboid or rectangular prism be more correct?

Likewise, in the text it is stated " can be treated as a polygon.", would it not be a polyhedron as it is 3D not 2D shape?

To expand the description of any pore shape or MOF type, a more general convention is used to sub divide the unit cell into five tetrahedra. It would be useful to show in Fig 2 the four other tetrahedra which describe the cell and the four vertices O, A, B, C should be defined on Fig 2.

For the size of cells, I wonder why an equal number of cells were used, when the closing direction is perpendicular to the dabco axis, and therefore could lower the computational cost of the simulations or allow for larger crystal sizes in the other directions?

Can the authors comment on the force constant testing for different sized systems? The force constants seem to follow no trend for the same system at different sizes, e.g. 0.002 kcal mol⁻¹ was chosen, which achieves similar ΔV for 3x3x3 and 4x4x4 but is an order of magnitude lower for 6x6x6. It also seems to give fluctuations of ΔV 1-10 Å³, not 100 Å³?

Could the authors comment on the periodicity of the pore closed structure of DMOF-1 formed in the mechanism? The authors state that the cp form is energetically unfavourable compared to that of the cp DUT-128. To date there is no experimental structure of a cp DMOF-1 structure, I was wondering if the authors could further infer from their simulations why this might be?

Additionally, would it be possible from this method of controlling the volume of the NC to determine the pressure of closing?

For the DUT-128, the size of the cell gave some discrepancies to the pore closing mechanism. Highlighted in the 4x4x4 case. Could the authors comment further on the random nature of the 4x4x4 case, could this be due to the larger linker in DUT-128 and allowing metastable states to more easily exist, as mirrored in the free energy curves in Fig 9?

Once these concerns are taken into account I would be happy to accept this manuscript to Communications Chemistry, as I see this as a good step forward for the MOF simulation community.

Response to the Reviewer's Comments

We would like to thank all the reviewers for their time and effort reading our manuscript, and their valuable comments and criticism. We have addressed all the points, raised by the reviewers and respond to them inline in the following (in blue color, with additions to the manuscript indicated in a red font):

Reviewer #1

In „Simulating the Breathing Phase Transitions of Metal-organic Frameworks : Volume Control of Nanocrystallites” Schaper and Schmid continue previous molecular dynamics (MD) investigations on framework flexibility on finite-sized crystals of metal-organic. The current work addresses the well-known DMOF-1 structure as a starting model and applies the methodology of direct volume control. The methodology is then applied to the recently reported framework material with extended ligand length named DUT-128. The application of direct volume control provides full access to the phase change and processes occurring upon framework deformation as well as the underlying free energy landscapes. This allows to sample nanocrystals of MOFs without periodic boundary conditions up to volumes of 1400000 Å³ (6x6x6 supercells). The manuscript is well written and illustrated. I particularly enjoyed the discussion on comparing the various computational approaches with respect to experimental procedures. The recent experimental observations of such finite crystal size phenomena can now finally be accessed computationally with even higher accuracy and lower costs which is a big step forward – Congratulations to the authors! As far as I can tell the manuscript and ESI contains all relevant information (github repository Link will be provided when submitting a final manuscript for publication.) to reproduce the study and potentially extend it to other systems. As such I strongly recommend publication of this manuscript and in the following provide a few suggestions and questions that might improve the manuscript even further.

We would like to thank the reviewer for his very positive assessment of our work.

1. *I suggest to add a figure to extend Fig 6 and 9 or a new panel with normalized (to a certain volume/energy) energy landscapes and directly compare as an overlay the shape of the free energy landscape upon increasing the size of the NCs.*

We agree with the reviewer, that a normalized comparison of the free energy curves would be desirable, but especially for such still-small NCs, it is not exactly clear on which basis this could be done. One possibility would be to normalize with respect to the number of paddle-wheel units in the system. Another option is to refer to the number of pores. We have added two figures in the SI (Figure 3 and Figure 4) using the latter approach for both DMOF-1 and DUT-128, indicating that the free energy difference between cp and op phase is roughly the same, but there is no clear tendency observable concerning the NC's size. In contrast, even and odd sizes seem to differ since DMOF-1 4x4x4 and 6x6x6 show a somewhat lower free energy difference than 3x3x3 and 5x5x5. In addition, it must be pointed out that the unnormalized total free energies of the entire NC, especially for the barriers, will determine the system's kinetic behavior as for a large molecular system. Note that the free energy barriers are substantially smaller than the corresponding barrier for a concerted process under PBC (see ref. 14, Fig. 10).

We added a brief statement on the barriers in the manuscript:

“First, for all sizes, the phase transformation implies overcoming a free energy barrier between the open and the completely closed pore form, which is substantially smaller than for the PBC case [14, 34] after normalizing per pore (see SI).”

and added two figures with the normalized free energies and a short paragraph, explaining the observations, to the Supporting Information.

2. *Fig.8: a) and b) should be defined*

We thank the reviewer for spotting this. The Figures are now mentioned in the manuscript.

3. *Have the authors considered extracting information about the nature and impact of volume to surface ratios in this study? Is there a way to extract such information in particular on the formation of metastable states?*

We thank the reviewer for this suggestion. Yes, it is possible to determine the external surface of the tessellated representation of the NC's volume by summing up all faces of the tetrahedrons that face outside of the NC. This would give access to an instantaneous approximation of the surface-to-volume ratio of the NC during the simulation. We will follow this suggestion and implement this in our code as an additional tool for analysis. However, we have not done this in the current investigation since the volume changes primarily dominate the variation of the surface-to-volume ratio. Therefore, the total volume will serve equally well as a measure of the NCs state. Note that the significant changes in the surface-to-volume ratio are due to the change in the

NC's size.

4. *An open question in the community of flexible MOFs is the correlation of the activation energy for a transition determined with PBCs (usually per unit cell) to the activation barrier in a finite crystal. Do the authors see a way to extrapolate the current findings towards even larger super cells without actually having to perform the MD simulations?*

We fully agree here with the reviewer and refer here also to point 1. above: The free energy barrier for a phase transition for a finite system (which is equivalent to the “real system” as opposed to PBC, which is always an approximation) must be considered as the total (unnormalized) barrier for the process. Already for the still small sizes of NCs, investigated in this work, a substantially smaller barrier is observed compared to the PBC case. This barrier will converge to a limiting value even for larger NCs since only a (small) fraction of the entire system needs to undergo the deformation for the phase transition to occur. This will indeed allow us to extrapolate the limiting barrier (within the accuracy of the theoretical model) for arbitrarily large systems. We believe, however, that further simulations on larger NCs will be necessary to confirm that this convergence has been reached. We are currently working on this in our group.

To clarify this aspect we added the following sentences to the manuscript: “For even larger NCs, the barrier is expected to converge to a particular value since only a limited fraction of the pores will have to undergo a structural transition for the pore closing. This will allow us to extrapolate the limiting barrier for arbitrarily large systems.”

5. *Can this study be conducted with guest molecules included in the pores and would that be an accurate way of sampling guest-induced transitions similar to MD-GCMC methods?*

Yes, this is possible, and we are currently working on this. Note that in PBC simulations, the adsorption of guest molecules implies a Grand Canonical approach since there is no interface to the “gas reservoir” in the system. Thus, in the mentioned MD-GDMC methods, guest molecules “appear” or “vanish” within the pores of the MOF. For an NC simulation, there is, however, an explicit interface, and guest molecules can enter the MOF through the surface or leave it. Thus, such a simulation includes both potential surface effects (given that the surface is adequately described) and transport. The downside is that sufficiently long sampling times will be necessary to equilibrate the system properly.

We added the following sentence in the Conclusions Section of the manuscript:

“In addition, the method could be combined with the explicit simulation of guest molecules surrounding the NC, which allows to study transport and adsorption in a joint fashion.”

Reviewer #2

This manuscript reports on molecular dynamics simulations of structural transformations of MOF frameworks. The authors have been working on this topic using computer simulations, and this manuscript is one of them. The focus of this study is to improve a simulation technique and solve some problems confirmed in the pressure bath concept that the authors previously proposed. In this paper, the authors proposed the volume control method and successfully reproduced structural transformation processes from open pore form to closed pore form. However, the results obtained in this study basically followed those obtained from previous simulations using the pressure bath concept. In that sense, the novelty of this study is doubtful from a viewpoint of MOF chemistry and materials. Thus, this manuscript is not suitable for the publication in Communications Chemistry, and rather more suitable for a journal dealing with simulation techniques and theoretical approaches. Another comment is on the title of this manuscript. Considering the present results in which guest molecules are not treated in the simulations, "breathing phase transitions" should be replaced with "structural transitions" because breathing is accompanied by host framework transformation and guest molecule adsorption/desorption.

We thank the reviewer for his critical words. In particular, we have to admit that the use of the term "breathing phase transitions" is misleading, even though it is often used to refer, in general, to the volume-changing structural transitions of MOFs. Thus, we have replaced the term "breathing phase transition" with "structural transition" in the title and throughout the entire manuscript. However, we have to disagree with the reviewer's point of view that our work is just a continuation of the previous work on the pressure bath. To our knowledge, all theoretical simulations of MOF phase transitions have been performed in PBC, with the great majority using rather small unit cells. We are convinced that investigating finite-sized NCs with an explicit surface will be a paradigm change in this field, and the necessary methodology to do this needs to be developed, which is of relevance for the entire community. The previously proposed method of a pressure bath was a step in the right direction. However, it did not allow for the determination of free energies along the path of transformation but just allowed to exert of a trigger. Thus, the proper volume control of arbitrarily shaped NCs is inevitable to investigate the volume-changing structural transitions of MOF NCs in the same way it is routinely done for PBCs. This is what we could demonstrate in this work. Since the special issue of Communications Chemistry" is entitled "Modelling and Advanced Characterization of Framework Materials," we strongly believe that our contribution is suitable.

Reviewer #3

The authors have produced an interesting piece of work on modelling the breathing behaviour of metal-organic frameworks via finite sized simulations of nano crystallite models. I believe this in an underdeveloped area in the field, firstly that the periodic models used to simulate these effects rarely agree with experimental observations, and finite sized models have only been recently introduced into the literature in the past 3-4 years. Therefore I welcome this new approach. The authors begin with outlining how their new approach to modelling finite sized systems via tetrahedra and calculate the volume and derivative.

We would like to thank the reviewer for his very encouraging words.

1. *Here I have a few questions relating to the terminology describing the shape of DMOF-1. DMOF-1 is tetragonal in symmetry, the initial assumption in the text of Fig 2 it is stated " can be treated as a cube defined by the eight paddle-wheel vertices", would it not be treated as a cuboid or rectangular prism be more correct? Likewise, in the text it is stated " can be treated as a polygon.", would it not be a polyhedron as it is 3D not 2D shape?*

The reviewer is absolutely correct, and we have corrected these errors. As suggested by the reviewer, we used the term “cuboid” for the pore of the tetragonal DMOF-1.

2. *To expand the description of any pore shape or MOF type, a more general convention is used to sub divide the unit cell into five tetrahedra. It would be useful to show in Fig 2 the four other tetrahedra which describe the cell and the four vertices O, A, B, C should be defined on Fig 2. For the size of cells, I wonder why an equal number of cells were used, when the closing direction is perpendicular to the dabco axis, and therefore could lower the computational cost of the simulations or allow for larger crystal sizes in the other directions?*

We can see the point the reviewer is making here. However, the four other tetrahedra are actually shown in Fig 2., since they are formed by one face of the central (orange or blue) tetrahedron and the black lines of the surrounding cube. This is also explained in the text. “There is one tetrahedron inside the pore (colored in blue and orange, respectively) and four tetrahedrons built by the sides of the pore and one of the faces of the centered tetrahedron.” In order to make this aspect more transparent for the reader, we extended Fig. 2 (including the caption), showing the five tetrahedrons representing the cuboid explicitly, and added a part to explain the vertex labeling. In fact, the choice of which vertex is O, A, B, or C is entirely arbitrary, and this choice must be made once in the beginning.

In order to clarify this point, we have rephrased the description of how the volume is

computed. The sentence now reads:

The volume of one tetrahedron with the vertices O, A, B, and C can be calculated by

$$\begin{aligned} V_{tet} &= \frac{1}{6} \cdot \left| \vec{a} \cdot (\vec{b} \times \vec{c}) \right| \\ &= \frac{1}{6} \cdot |a_x \cdot (b_y \cdot c_z - b_z \cdot c_y) + a_y \cdot (b_z \cdot c_x - b_x \cdot c_z) + a_z \cdot (b_x \cdot c_y - b_y \cdot c_x)| \end{aligned} \quad (1)$$

with the vectors \vec{a} , \vec{b} , and \vec{c} , pointing from O to A, B, and C, respectively. Note, that labeling the vertices is arbitrary and must be done once in the beginning for each of the five tetrahedrons, forming one of the two representations.

In his last point, the reviewer refers to the question of using different aspect ratios for the NCs. The reviewer is absolutely correct that unequal aspect ratios with the c-axis (pillar) being shorter than a and b could be used. We are currently working to perform calculations with non-unit aspect ratios, including the opposite, namely rod-like NCs with an extended c-axis, to study how the phase transitions travel in this direction. However, in the current work, we wanted to stay as close as possible to the previous investigations using a mechanical force and the pressure bath to trigger the phase transition (refs. 14 and 18), where such unit aspect ratios were used.

3. *Can the authors comment on the force constant testing for different sized systems? The force constants seem to follow no trend for the same system at different sizes, e.g. 0.002 kcal mol⁻¹ was chosen, which achieves similar ΔV for 3x3x3 and 4x4x4 but is an order of magnitude lower for 6x6x6. It also seems to give fluctuations of ΔV 1-10 A³, not 100 A³?*

The reviewer is absolutely right, and we are very grateful for pointing us to this partly erroneous data. First of all, we intended to identify the range of values for k where the constraint is sufficiently well maintained during the SMD simulations but at the same time gives enough freedom to allow for volume fluctuations, which is especially important for the US simulations, where the umbrella windows need to overlap sufficiently. Due to a mistake, the original values for the fluctuations were incorrect. This is now corrected, and we present the standard deviation in the volume to indicate the width of the fluctuations. Furthermore, the test system was initiated from an MD simulation in the volume range of the open pore phase. However, for the different systems, 3x3x3, 4x4x4, and 6x6x6, a random starting structure was chosen, and thus, there is no size trend visible, as the reviewer correctly points out.

All in all, since this screening of the constraint force constant is of minor relevance, we decided to shift this part into the Supporting Information as a new Section 1.1 "Force Constants". A short part explaining the procedure has been moved to the Computational

details Section of the paper and reads as follows:

“To apply the volume restraining external potential, a proper choice of the force constant k is needed since it controls the strength of the potential to bias the volume. For this purpose, a screening experiment was performed using DMOF-1 NCs in the op form as a test system in order to determine its magnitude. Further details are given in the Supporting Information. As a result, a force constant $k = 0.002 \text{ kcal} \cdot \text{mol}^{-1} \cdot \text{\AA}^{-6}$ was used in all further SMD calculations to maintain a difference between the mean volume of the respective NC during the simulation and the reference volume of approximately $\Delta V = 10 \text{ \AA}^3$ up to $\Delta V = 80 \text{ \AA}^3$ for all investigated sizes, while the fluctuation of the volume is $\Delta\Delta V \geq \pm 50 \text{ \AA}^3$.”

In addition, we have slightly rearranged the Computational Details Section and added a short paragraph on implementing the volume control constraint, which we believe is helpful for the reader:

“The volume constraint has been implemented in Python using the LAMMPS fix python/invoke in order to perform calculations of the NC’s volume and the forces resulting from the external potential U_{ext} (Eq. 3) in each timestep. For numerical efficiency, the computation is done in parallel, distributing over the pores, and the core routines are accelerated by a just-in-time compiler Numba [25]”

4. *Could the authors comment on the periodicity of the pore closed structure of DMOF-1 formed in the mechanism? The authors state that the cp form is energetically unfavourable compared to that of the cp DUT-128. To date there is no experimental structure of a cp DMOF-1 structure, I was wondering if the authors could further infer from their simulations why this might be?*

The reviewer is correct that DMOF-1 does experimentally not form a cp form. In previous work, we investigated DMOF-1 by in-situ PXRD under hydrostatic pressure (ref 34), where some indications for a phase transformation could be observed before the system amorphizes. In this work, we used the same force field for DMOF-1 in PBC and could observe a well-ordered cp phase at higher pressures. Note, however, that this is because our force field is non-reactive, and no bonds can be broken. Thus our theoretical model is not able to describe any kind of bond-breaking effects. All we can say is that under the constraint of not allowing any bond breaking, an ordered cp phase could be formed, which is substantially destabilized concerning the op phase. For DUT-128 with the increased dispersive interaction, in contrast, the cp phase is stabilized, and indeed, one can observe such a system also experimentally.

This was already mentioned in Sec. 3.2.1 of the original manuscript, but we have extended and rephrased this part as follows:

“Experimentally, DMOF-1 is a non-flexible MOF, which tends to amorphize under

hydrostatic pressure [34]. However, bond breaking can not be simulated for a non-reactive model like MOF-FF, and a fictitious stable but high-energy closed pore form is observed in the simulations.”

5. *Additionally, would it be possible from this method of controlling the volume of the NC to determine the pressure of closing?*

The difficulty is actually that pressure is exerted by gas molecules, which are currently not present in our simulation. Thus, the concept of “pressure” is also not really present in the current setting. At the same time, these guest molecules will adsorb within the NC and alter its phase transition behavior. However, as already mentioned in response to point 5 of reviewer #1, it is possible to include gas molecules in the simulation and to compute the free energy wrt to volume curves at different gas phase pressures. This will allow us to determine the limiting pressures where barriers for transitions for op \rightarrow cp (or reverse) will become small enough for the system to switch.

We have extended a sentence in the Concluding Section to reflect this aspect as follows:

“Its application to MOF NCs will lead to a more detailed understanding of the underlying mechanism, i.e., to elucidate the surface and transport effects of guest molecules inducing structural transition processes by determining the limiting pressure of closing, which can be used to improve the applications of flexible materials by the targeted design of the phase transition behavior.”

6. *For the DUT-128, the size of the cell gave some discrepancies to the pore closing mechanism. Highlighted in the 4x4x4 case. Could the authors comment further on the random nature of the 4x4x4 case, could this be due to the larger linker in DUT-128 and allowing metastable states to more easily exist, as mirrored in the free energy curves in Fig 9?*

The reviewer is correct that the choice of the 4x4x4 to highlight the case was randomly chosen. Our central point was to show that the volume control leads to a reaction coordinate that does not enforce a specific mechanism, as in our prior work (Ref. 14), where we used a mechanical force pulling the edges of the NCs. However, for sufficient sampling time and several trajectories, we expect the energetically most stable diamondoid cp form to be formed in the end. Nevertheless, the reviewer is probably right that the extended linker in DUT-128, with its higher dispersive interaction in the cp form, could lead to “locked-in” metastable states. For larger NCs, it can be expected that domains with different folding directions could form, which must be considered a metastable form of the corresponding all-ordered cp system.

We have rephrased and extended the part at the end of Section 3.3 as follows:

“An exception is observed in the trajectories for the 4x4x4 NC: Figure 8b reveals that

the partially closed pore form differs in one case. Here, no distinct phase boundary can be identified, and the pore closing starts randomly in several pores, resulting in a non-diamond-shaped form. This demonstrates that the volume control method does not enforce a specific mechanism and allows to explore the potential energy landscape of the structural change mechanism in an unbiased way.”

Once these concerns are taken into account I would be happy to accept this manuscript to Communications Chemistry, as I see this as a good step forward for the MOF simulation community.

REVIEWERS' COMMENTS:

Reviewer #1 (Remarks to the Author):

I thank the authors for the detailed and extensive revision. They have addressed all the points raised accurately and refined both the manuscripts and the presented data. I am sure that this work will receive attention by the wider community and I look forward to see the authors address the future simulations as discussed in the rebuttal letter.

Reviewer #3 (Remarks to the Author):

The authors have comprehensively addressed all comments and queries I had on the manuscript. I am happy for the manuscript to be accepted in its current state.